# Six New Phenylpropanoid Derivatives from Chemically Converted Extract of *Alpinia galanga* (L.) and Their Antiparasitic Activities

**DOI:** 10.3390/molecules26061756

**Published:** 2021-03-21

**Authors:** Melanny Ika Sulistyowaty, Nguyen Hoang Uyen, Keisuke Suganuma, Ben-Yeddy Abel Chitama, Kazuhide Yahata, Osamu Kaneko, Sachiko Sugimoto, Yoshi Yamano, Susumu Kawakami, Hideaki Otsuka, Katsuyoshi Matsunami

**Affiliations:** 1Graduate School of Biomedical and Health Sciences, Hiroshima University, 1-2-3 Kasumi, Minami-ku, Hiroshima 734-8553, Japan; melanny-i-s@ff.unair.ac.id (M.I.S.); kimdonguyen452@yahoo.com (N.H.U.); ssugimot@hiroshima-u.ac.jp (S.S.); yamano@hiroshima-u.ac.jp (Y.Y.); 2Faculty of Pharmacy, Universitas Airlangga, Surabaya 60286, Indonesia; 3National Research Center for Protozoan Diseases, Obihiro University of Agriculture and Veterinary Medicine, Inada, Obihiro 080-8555, Hokkaido, Japan; k.suganuma@obihiro.ac.jp; 4Department of Protozoology, Institute of Tropical Medicine (NEKKEN), Nagasaki University, 1-12-4 Sakamoto, Nagasaki 852-8523, Japan; benchitama@gmail.com (B.-Y.A.C.); kyahata@nagasaki-u.ac.jp (K.Y.); okaneko@nagasaki-u.ac.jp (O.K.); 5Department of Natural Products Chemistry, Faculty of Pharmacy, Yasuda Women’s University, 6-13-1 Yasuhigashi, Asaminami-ku, Hiroshima 731-0153, Japan; kawakami@yasuda-u.ac.jp (S.K.); otsuka-h@yasuda-u.ac.jp (H.O.)

**Keywords:** *Alpinia galanga*, *Leishmania*, *Trypanosoma*, Plasmodium, chemically converted extract, unnatural natural product, phenylpropanoid, malaria, NTDs (neglected tropical diseases)

## Abstract

Chemical conversion of the extract of natural resources is a very attractive way to expand the chemical space to discover bioactive compounds. In order to search for new medicines to treat parasitic diseases that cause high morbidity and mortality in affected countries in the world, the ethyl acetate extract from the rhizome of *Alpinia galanga* (L.) has been chemically converted by epoxidation using dioxirane generated in situ. The biological activity of chemically converted extract (CCE) of *A. galanga* (L.) significantly increased the activity against *Leishmania major* up to 82.6 ± 6.2 % at 25 μg/mL (whereas 2.7 ± 0.8% for the original extract). By bioassay-guided fractionation, new phenylpropanoids (**1**–**6**) and four known compounds, hydroquinone (**7**), 4-hydroxy(4-hydroxyphenyl)methoxy)benzaldehyde (**8**), isocoumarin *cis* 4-hydroxymelein (**9**), and (2*S*,3*S*,6*R*,7*R*,9*S*,10*S)-*humulene triepoxide (**10**) were isolated from CCE. The structures of isolated compounds were determined by spectroscopic analyses of 1D and 2D NMR, IR, and MS spectra. The most active compound was hydroquinone (**7**) with IC_50_ = 0.37 ± 1.37 μg/mL as a substantial active principle of CCE. In addition, the new phenylpropanoid **2** (IC_50_ = 27.8 ± 0.34 μg/mL) also showed significant activity against *L. major* compared to the positive control miltefosine (IC_50_ = 7.47 ± 0.3 μg/mL). The activities of the isolated compounds were also evaluated against *Plasmodium falciparum*, *Trypanosoma bruce*i *gambisense* and *Trypanosoma bruce*i *rhodeisense.* Interestingly, compound **2** was selectively active against trypanosomes with potent activity. To the best of our knowledge, this is the first report on the bioactive “unnatural” natural products from the crude extract of *A. galanga* (L.) by chemical conversion and on its activities against causal pathogens of leishmaniasis, trypanosomiasis, and malaria.

## 1. Introduction

*Plasmodium*, *Leishmania*, and *Trypanosoma* are causal microbes of malaria and neglected tropical diseases (NTDs) and are transmitted by certain species of insects as vector-borne diseases. Malaria is a life-threatening disease caused by *Plasmodium* parasites. The World Health Organization (WHO) estimated 229 million cases and 409,000 deaths by malaria in 2019. Leishmaniases are grouped into three forms, visceral (kala-azar), cutaneous and mucocutaneous. The most common leishmaniasis is cutaneous leishmaniasis caused by *Leishmania major* and so on. It is estimated that 700,000 to 1 million new cases and 26,000 to 65,000 deaths occur annually [1,2]. In 2018, 977 cases were reported for human African trypanosomiasis (sleeping sickness), which takes two forms, depending on the parasite involved, *T. b. gambiense* or *T. b. rhodesiense*, and affects the central nervous system [3].

Natural products have been recognized as an important resource in drug discovery and development due to their structural uniqueness and relatively strong biological activities. Newman and Cragg emphasized that natural product and natural product structures continue to play a highly significant role in drug discovery and development. For instance, of the 175 small molecules approved as anticancer agents from 1981 to 2014, 131 (75%) were natural or natural products-related compounds. Therefore, natural products and their derivatives are still playing an essential role in the discovery of drug candidates [4,5].

Natural products chemists have already discovered more than 313,000 natural products as listed in The Dictionary of Natural products so far [6]. Recently, an issue is evoked that the discovery of novel bioactive compounds is becoming difficult gradually. In the last decade or so, an attractive approach has been introduced for generating bioactive compounds having chemical diversity via chemical modification of crude natural extracts. This approach is an obvious method for altering the chemical composition of crude herbal extracts, which causes changes in the chemical structure and biological activity in a negative or positive manner. For example, sulfonylation with *p*-toluene sulfonyl chloride [7,8], ammonolysis with hydrazine monohydrate [9,10,11], bromination with bromine [8,11], oxidation with Dess–Martin periodinane (DMP) [12], reduction with sodium borohydride [12], epoxidation with *m*-chloroperbenzoic acid (*m*CPBA) [12] and methyl(trifluoromethyl)dioxirane followed by transannulation with a Lewis acid catalyst [13], and ethanolysis with hydroxylamine hydrochloride [14] to natural extracts have been carried out to produce novel or bioactive compounds.

*Alpinia galanga* (L.) belongs to a family Zingiberaceae, which is widely distributed in China, India, and Southeast Asian countries, including Indonesia. The rhizome, fruit, and flowers of this plant have been used as flavoring agents, food additives, or gastroprotective agents. In addition, it is traditionally applied for treating dyspepsia and abdominal colic pain, motion sickness, and gastralgia [15,16]. Up to now, there are numerous articles about pharmacological activities of *A. galanga*, for instance antiallergic [17], anticancer [18,19,20], anti-inflammatory [21], antimicrobial [21,22,23,24], antioxidant [24,25], anti-amnesiac [26], antiasthma [27], and also melanogenesis inhibitor [28]. However, there is no report about its positive activity as an antiparasitic agent [29]. Thus, *A. galanga* has already been studied extensively, including chemical constituents and biological activities, and it is unlikely that any further progress will be obtained easily by the standard method. Therefore, we applied the chemical derivatization method to generate new bioactive compounds.

In this report, a chemical diversification of *A. galanga* extract resulted in a generation of bioactivity against the *Leishmania* parasite, and new (**1**–**6**) and bioactive compounds were isolated through bioactivity-guided fractionation. The chemical structures were elucidated by intensive spectroscopic analyses. Activities against leishmaniasis, human African trypanosomiasis, and malaria are also discussed.

## 2. Results and Discussion

### 2.1. Isolation of Chemical Components of Chemically Converted Extract of Alpinia Galanga

The dried rhizomes of *A. galanga* (2.0 kg) were extracted with MeOH (3 × 10 L) at room temperature to give MeOH extract. Afterward, this extract was partitioned by solvent fractionation with ethyl acetate (EtOAc)/H_2_O to give EtOAc extract (E). The EtOAc extract was treated with dioxirane to produce the chemically converted extract (CCE) (Scheme 1). Dioxirane is a powerful reagent for epoxidation, which is generated in situ by the reaction of potassium peroxomonosulfate with acetone [30]. After being quenched and evaporated, HPLC analyses and biological assays of CCE and E were carried out. The anti-*Leishmania* activity of E and CCE against promastigotes *of L. major* was conducted by the colorimetric MTT (3-(4,5-dimethylthiazol-2-yl)-2,5-diphenyl-tetrazolium bromide) method [31]. The HPLC profile showed that major peaks in the original extract have vanished, and some new peaks appeared in the CCE (Figure 1). This result exhibited that new chemical components were produced by the chemical treatment. Furthermore, the inhibitory activity of CCE was significantly increased after treating with dioxirane. The CCE demonstrated 82.6 ± 6.2 % of inhibition against *L. major*, while the original EtOAc extract did not inhibit (2.7 ± 0.8 %) at 25 μg/mL (Figure 1).

Subsequently, by using the bioassay-guided isolation procedure, the CCE was further separated by various column chromatographies and HPLC to provide new compounds **1**–**6** (Scheme 1). In addition, four known compounds were identified such as hydroquinone (**7**) [32], 4-hydroxy(4-hydroxyphenyl)methoxy)benzaldehyde (**8**) [33], isocoumarin *cis* 4-hydroxymelein (**9**) [34], and (*2S*,*3S*,*6R*,*7R*,*9S*,*10S)-*humulene triepoxide (**10**) [35] by comparing with the spectral data (see Appendix A).

Compound **1** was obtained as a colorless liquid having molecular formula C_10_H_13_O_3_Cl from an [M + Na]^+^ ion at *m*/*z* 239.0446 with characteristic 3: 1 pattern of isotopic ^35^Cl and ^37^Cl in the high resolution electrospray ionization (HR-ESI) MS. The IR data exhibited absorption bands for hydroxy groups (3351 cm^−1^) and the C-Cl stretching bond (850 cm^−1^). The ^1^H NMR data (Table 1) showed the presence of one methoxy group at 3.36 ppm and a para-disubstituted aromatic ring with ortho-coupled protons at 7.21 and 6.76 ppm as doublets *(J* = 8.6 Hz) each in **1**. The ^1^H and ^13^C-NMR data were closely similar to those of the analogous compound, *erythro*-2-chloro-1-(4-hydroxyphenyl)propane-1,3-diol. On the basis of the coupling constant of H-1′ and H-2′ (*J* = 5.9 Hz), **1** was assigned as *erythro* in accordance with a reference (*J* = 5.9 Hz) [36]. Therefore, the structure of **1** was determined to be 4-[*threo*-2-chloro-1-hydroxy-3-methoxypropyl]phenol.

Compound **2** was obtained as a colorless liquid. Its molecular formula showed to be C_11_H_16_O_3_ from a pseudo molecular ion at *m*/*z* 219.0991 ([M + Na]^+^) in the HRESIMS. The IR data showed bands for a hydroxyl group (3379 cm^−1^) and an aromatic ring (1604, 1510, and 1442 cm^−1^). The ^1^H-NMR data (Table 1) showed two methoxy groups at 3.14 and 3.30 ppm, and a *para*-substituted aromatic ring at 7.10 and 6.77 ppm as doublets *(J* = 8.5 Hz). According to the correlation spectroscopy (COSY) and heteronuclear multiple bond connectivity correlations (HMBC) (Figure 2), **2** was assigned as 4-(1,3-dimethoxypropyl)phenol.

Compounds **3** and **4** were isolated as different peaks by HPLC. Compounds **3** and **4** were colorless liquids with the same molecular formula C_10_H_14_O_4_ by HRESIMS at *m/z* 221.0781 ([M + Na]^+^). The IR data indicated the presence of hydroxy groups (**3**: 3384, and **4**: 3394 cm^−1^) and an aromatic ring (**3**: 1606, 1514, and 1455, and **4**: 1616, 1514, and 1455 cm^−1^). The ^1^H NMR data (Table 1) displayed signals caused by the presence of one methoxy group (**3**: 3.26, and **4**: 3.34 ppm), a para-substituted benzene ring at δ_H_ 6.74 (2H, d, *J* = 8.5 Hz) and 7.16 (2H, d, *J* = 8.5 Hz) for **3**, and δ_H_ 6.76 ppm (2H, d, *J* = 8.6 Hz) and 7.21 (2H, d, *J* = 8.6 Hz) for **4**. The ^13^C-NMR data (Table 2) exhibited eight carbon peaks that were classified by DEPT and heteronuclear single quantum coherence (HSQC) spectrum as one methoxy carbon group (**3**: δ_C_ 59.2, and **4**: δ_C_ 59.3), two *sp^2^* methine carbons (**3**: δ_C_ 116.0 (2C), 129.1(2C), and **4**: δ_C_ 115.8 (2C), 129.4(2C)), two *sp^2^*quaternary carbons (**3**: δ_C_ 133.9 and 158.0, and **4**: δ_C_ 133.9 and 157.9) and three oxygenated carbons (**3**: δ_C_ 74.9, 75.5, and 76.0, and **4**: δ_C_ 75.1, 75.0, and 75.6). These results indicated that compounds **3** and **4** have the same planar structure as shown in Scheme 1. Yang et al. and Li et al. reported slight but practical differences between *erythro-* and *threo-* forms of the related compounds, 1-(4-hydroxyphenyl)-1-methoxy-2,3-propanediol: that is, the *erythro*- and *threo*- forms had different coupling constants of H-1′ (6.6 Hz and 6.3 Hz, respectively), and the *erythro*-form showed a 0.9 ppm higher-field chemical shift value at C-2′ compared to that of *threo*-form. In the ^1^H and ^13^C-NMR spectrum of **3** and **4**, the coupling constants of H-1′ were 6.6 and 6.3 Hz, respectively, and the relatively lower-field chemical shift value at C-2′ of **3** (δ_C_ 76.0) compared to C-2′ of **4** (δ_C_ 75.0) suggested *threo-* and *erythro*-form for **3** and **4**, respectively, as shown in Scheme 1 [37,38].

Compound **5** was yielded as a colorless liquid with molecular formula C_11_H_16_O_4_ determined by HRESIMS from the peak at *m/z* 235.0939 ([M + Na]^+^). The IR data indicated bands for a hydroxy group (3379 cm^−1^) and an aromatic ring (1604, 1509, and 1455 cm^−1^). The ^1^H NMR data (Table 1) showed peaks caused by two methoxy groups at 3.19 and 3.24 ppm, and a para-substituted coupling system at δ_H_ 6.78 (2H, d, *J*= 8.5 Hz) and 7.13 (2H, d, *J*= 8.5 Hz). The ^13^C-NMR data (Table 2) exhibited nine carbon peaks that were classified by chemical shift values, DEPT and HSQC spectrum as two methoxy carbons (δ_C_ 56.8 and 59.2), two *sp2* methine carbons (δ_C_ 116.2 (2C), 129.9 (2C)), two *sp^2^* quaternary carbons (δ_C_ 130.6 and 158.5) and three oxygenated carbons (δ_C_ 74.6, 75.6, and 85.7). Chakraborty et al. recently stereoselectively synthesized the related compounds, *threo*- and *erythro*-1-(4-hydroxyphenyl)-1-methoxy-2,3-propanediol, and the coupling constants of H-1′ of *threo*- and *erythro*- forms were 7.0 and 6.4 Hz, respectively. According to this report, compound **5** showing the coupling constant of H-1′ (*J* = 7.3 Hz) can be concluded as a *threo* isomer, 4-[*threo*-2-hydroxy-1,3-dimethoxypropyl]phenol [39].

Compound **6** was isolated as a colorless liquid with molecular formula C_13_H_18_O_4_ as measured by HRESIMS at *m/z* 261.1099 ([M + Na]^+^). The IR data indicated absorption bands for a hydroxyl group (3368 cm^−1^) and the presence of an aromatic ring (1607 and 1512 cm^−1^). The ^1^H NMR data (Table 1) showed peaks caused by one methoxy group at 3.35 ppm, two methyl group at 1.46 and 1.51 ppm, and a para-disubstituted coupling system at δ_H_ 6.79 ppm (2H, d, *J*= 8.6 Hz) and 7.21 (2H, d, *J*= 8.6 Hz). The ^13^C-NMR data (Table 2) exhibited 11 carbon peaks that were categorized as two methyl carbons group (δ_C_ 27.3 and 27.4), one methoxy group (δ_C_ 59.6), two *sp2* methine carbons (δ_C_ 116.4 (2C) and 129.3(2C)), two *sp^2^* quaternary carbons (δ_C_ 129.5 and 158.8), and three oxygenated carbons (δ_C_ 72.6, 80.9, and 83.8). These results suggested a planar structure of **6** as shown in Scheme 1. The related acetonide, 4-(hydroxymethyl)-5-[3‘,5‘-dihydroxyphenyl]-2,2-dimethyl-1,3-dioxolane, a *threo* isomer, has coupling constant *J*= 8.1 Hz for H-1′ and H-2′ [40]. Morikawa et al. explained that *threo*- and *erythro*- forms of acetonides have similar chemical shift values except for the coupling constants of H-1′ and H-2′, that is, *erythro*- form (*J* = 2.8 Hz) and *threo*-form (*J* = 10.1 Hz) [41]. Therefore, based on the coupling constants of H-1′ and H-2′ (*J* = 8.7 Hz), compound **6** can be assigned as *threo* isomer as 4-[*threo*-5-(methoxymethyl)-2,2-dimethyl-1,3-dioxolan-4-yl]phenol.

All of the isolated compounds did not show significant optical rotation values, probably because of the formation of the racemic mixture through the chemical derivatization.

### 2.2. Antiparasitic Activity Examinations of the Isolated Compounds

All the isolated compounds were investigated for their biological activity toward selected protozoan parasites. Table 3 shows IC_50_ values of isolated compounds against *L. major, T. b. gambisense, T. b. rhodesiense,* and *P. falciparum.* Among isolated compounds, hydroquinone (**7**) had the highest activity against parasites belonging to the family Trypanosomatidae. The leishmanicidal and trypanocidal activities of hydroquinone derivatives had been reported [41,42]. Our result of **7** further supports the importance of the hydroquinone motif as an important pharmacophore. Compounds **2**, **8** and **9** possessed significant activities against both *L. major* and *Trypanosoma* parasites. On the other hand, compound **1** had a strong activity on *Trypanosoma* parasites only. However, all isolated compounds showed no activity against *P. falciparum* (Table 3), probably because of the biological difference at the phylum level; Sarcomastigophora (*L. major*, *T. b. gambiense*, and *T. b. rhodesiense*) and Apicomplexa (*P. falciparum*). Alternatively, another possible reason could be the absence of a peroxide ring system which is important for the antimalarial activity of artemisinin [43]. Therefore, a chemical modification to introduce peroxide functions may be a promising method to produce novel antimalarial agents in the future.

To our understanding, there are no reports on the biological activity of 4-hydroxyphenylpropanols against *L. major*, *P. falciparum*, *T. brucei gambisense*, and *T. brucei rhodeisense*. Therefore, our study is the first report on the activities of 4-hydroxyphenylpropanol and related compounds against those parasites. Ropes et al. reported the derivatives of *p*-coumaric acid alkyl esters and their inhibitory activities against *Leishmania* and *Plasmodium* [44]. The presence of alkyl function on C-3′ (-OMe) and the relatively non-polar nature of **1** and **2** may be related to the higher activity among **1**–**6**.

## 3. Materials and Methods

### 3.1. General Methods

Optical rotations were determined on a P-1030 digital polarimeter (JASCO, Tokyo, Japan). IR spectra were measured on a FT-710 Fourier transform infrared spectrophotometer (Horiba, Kyoto, Japan), while UV spectra were taken on a V-520 UV/Vis spectrophotometer (JASCO, Tokyo, Japan). NMR data were obtained by an Ultrashield 600 spectrometer (Bruker, Ettlingen, Germany) with tetramethylsilane (TMS) as an internal standard. Positive ion HR-ESI-MS measurement was performed with the LTQ Orbitrap XL mass spectrometer (Thermo Fisher Scientific, Waltham, MA, USA). Column chromatography (CC) was carried out on silica gel 60 (E. Merck, Darmstadt, Germany), and octadecyl silica (ODS) gel (Cosmosil 75C18-OPN (NacalaiTesque, Kyoto, Japan; Φ = 35 mm, L = 350 mm). HPLC was performed on ODS gel (Inertsil ODS-3, GL-science, Φ = 6 mm, 250 mm, flow rate = 1.6 mL/min, GL Sciences, Tokyo, Japan), and the eluate was observed by refractive index detector with the RI-930 intelligent detector (JASCO, Tokyo, Japan) and a PU-1580 intelligent pump (JASCO, Tokyo, Japan). All chemicals were purchased from Wako Pure Chemical Industries, Ltd. (Osaka, Japan) and TCI (Tokyo, Japan).

### 3.2. Plant Material

Rhizome of *A. galanga* (L.) was purchased from Shinwa-bussan Co., Osaka, Japan, in 2014. A voucher specimen (Ay-AG2014-Shinwa) was deposited in the herbarium of the Department of Pharmacognosy, Graduate School of Biomedical and Health Sciences, Hiroshima University, Japan.

### 3.3. Preparation of Chemically Converted Extract of A. galanga and Isolation of Compounds ***1**–**10***

Rhizome of *A. galanga* (2.0 kg) was cut into small pieces and extracted by maceration at room temperature with MeOH (10 L) three times and then evaporated to give an MeOH extract (303 g). This residue was suspended in 1 L of saturated sodium carbonate solution and partitioned with EtOAc (1 L) three times to produce an EtOAc extract (68 g).

The EtOAc extract (14 g) was dissolved in the mixture of acetonitrile (360 mL) and water (360 mL). Then, acetone (10.35 mL), sodium bicarbonate (19.6 g) and potassium peroxosulphate (28.2 g) were added sequentially to the solution at 0 °C. The mixture was stirred for 3 h at 0 °C and then potassium peroxosulphate (28.2 g) was added again to the solution with stirring for another 3 h at 0 °C. The mixture was then allowed to rise naturally to room temperature with stirring for 24 h. The obtained reaction mixture was poured into water and extracted with ethyl acetate three times. The combined EOAc layer was washed with water, brine, dried over sodium sulfate, and evaporated to yield a residue, designated as CCE of *A. galanga* (10.5 g).

The CCE was separated on a silica gel (50 g) CC and eluted with an increasing amount of MeOH in CHCl_3_ (20:1, 10:1, 7:1, 5:1, 3:1, 1:1, and 100% MeOH, 500 mL each), generating nine fractions. Fraction CCE2 (1.1 g), CCE3 (1.4 g) and CCE4 (4.0 g) were subjected to ODS CC with a step gradient from 10% aq. MeOH to 100% MeOH, (400 mL each), leading to eight fractions. The residue of fraction CCE 2.1 (226 mg) was purified by HPLC using 20% aq. acetone to give **1** (40.1 mg), **7** (hydroquinone, 102 mg), and **8** (4-hydroxy(4-hydroxyphenyl)methoxy)benzaldehyde, 20.2 mg). The residue of fraction CCE 2.2 (79.5 mg) was purified by HPLC using 30% aq. MeOH to give **1** (1.9 mg), **9** (isocoumarin *cis*-4-hydroxymelein, 2.2 mg), **2** (2.4 mg), **6** (5.8 mg), and **10** ((2*S*, 3*S*, 6*R*, 7*R*, 9*S*, 10*S)-*humulene triepoxide, 2.8 mg). The residue of fraction CCE 3.1 (297.5 mg) was purified by HPLC using 20% aq. acetone to give **7** (70.1 mg), **3** (20.1 mg), **4** (6.0 mg), and **5** (12.5 mg) (Appendix A).

*4-[erytho-2-chloro-1-hydroxy-3-methoxypropyl]phenol* (**1**)

Colorless oil, IR υ_max_ (film) cm^−1^: 3351, 2935, 1668, 1607, 1454, 1377, 1236, 1118, 1038, 850; UV λ_max_ (MeOH) nm (log ε): 276 (2.58); ^1^H NMR (CD_3_OD, 600 MHz), see Table 1; ^13^C-NMR (CD_3_OD, 150 MHz), see Table 2. HRESIMS (positive ion) *m/z*: 239.0446 [M + Na]^+^ (calcd for C_10_H_13_O_3_ClNa: 239.0445).

*4-(1,3-dimethoxypropyl)phenol* (**2**)

Colorless oil, IR υ_max_ (film) cm^−1^: 3379, 2930, 2854, 1718, 1604, 1510, 1442, 1233, 1104; UV λ_max_ (MeOH) nm (log ε): 259 (2.57); ^1^H NMR (CD_3_OD, 600 MHz), see Table 1; ^13^C-NMR (CD_3_OD, 150 MHz), see Table 2. HRESIMS (positive ion) *m/z*: 219.0991 [M + Na]^+^ (calcd for C_11_H_16_O_3_Na: 219.0992).

*threo-1-(4-hydroxyphenyl)-3-methoxypropane-1,2-diol* (**3**)

Colorless oil, IR υ_max_ (film) cm^−1^: 3384, 2987, 2831, 1677, 1606, 1514, 1455, 1240, 1119; UV λ_max_ (MeOH) nm (log ε): 276 (2.96); ^1^H-NMR (CD_3_OD, 600 MHz), see Table 1; ^13^C-NMR (CD_3_OD, 150 MHz), see Table 2. HRESIMS (positive ion) *m/z*: 221.0781 [M + Na]^+^ (calcd for C_10_H_14_O_4_Na: 221.0784).

*erythro-1-(4-hydroxyphenyl)-3-methoxypropane-1,2-diol* (**4**)

Colorless oil, IR υ_max_ (film) cm^−1^: 3394, 2930, 2360, 1679, 1616, 1514, 1455, 1240; UV λ_max_ (MeOH) nm (log ε): 276 (2.33); ^1^H NMR (CD_3_OD, 600 MHz), see Table 1; ^13^C-NMR (CD_3_OD, 150 MHz), see Table 2. HRESIMS (positive ion) *m/z*: 221.0781 [M + Na]^+^ (calcd for C_10_H_14_O_4_Na: 221.0784).

*4-(threo-2-hydroxy-1,3-dimethoxypropyl)phenol* (**5**)

Colorless oil, IR υ_max_ (film) cm^−1^: 3379, 2930, 2360, 1604, 1509, 1455, 1375; UV λ_max_ (MeOH) nm (log ε): 279 (3.26); ^1^H NMR (CD_3_OD, 600 MHz), see Table 1; ^13^C-NMR (CD_3_OD, 150 MHz), see Table 2. HRESIMS (positive ion) *m/z*: 235.0939 [M + Na]^+^ (calcd for C_11_H_16_O_4_Na: 235.0941).

*4-(threo-5-(methoxymethyl)-2,2-dimethyl-1,3-dioxolan-4-yl)phenol* (**6**)

Colorless oil, IR υ_max_ (film) cm^−1^: 3368, 2986, 1732, 1607, 1512, 1375, 1230, 1165, 1078; UV λ_max_ (MeOH) nm (log ε): 276 (2.95); ^1^H NMR (CD_3_OD, 600 MHz), see Table 1; ^13^C-NMR (CD_3_OD, 150 MHz), see Table 2. HRESIMS (positive ion) *m/z*: 261.1099 [M + Na]^+^ (calcd for C_13_H_18_O_4_Na: 261.1097).

### 3.4. Biological Assay of Compounds ***1**–**10***

The leishmanicidal activities of the isolated compounds were evaluated using MTT colorimetric assay, as described by Asaumi et al. [31] MTT, 3-(4,5-dimethylthiazol-2-yl)-2,5-diphenyltetrazolium bromide, was reduced to form an insoluble purple precipitate, formazan, by cell metabolic activity, which is known to correlate to the viable cell number. In a 96-well plate, 1 µL of sample solutions in dimethylsulfoxide (DMSO) and *L. major* (1 × 10^5^ parasite/well) in 99 µL of medium were added to each well. It was then incubated for 72 h in a CO_2_ incubator at 25 °C. M-199 medium, supplemented with 10% heat-inactivated fetal bovine serum and 100 µg/mL of kanamycin, was used. After incubation, 100 µL of MTT solution was added to each well and the plates were incubated for another 8 h. The absorbance of the formazan solution in DMSO was recorded using a microplate reader at λ540 nm.

The trypanocidal activities of the isolated compounds were performed in 96-well plates as previously explained with slight modifications [45]. The amount of ATP was proportional to the number of metabolically active cells. This method utilized the enzymatic reaction of luciferase and luciferin that requires ATP to generate luminescence. In brief, each well contained 100 µL of parasite culture (1 × 10^4^ parasites/well) with serial dilutions of compounds. After incubation for 72 h at 37 °C under 5% CO_2_, 25 µL of CellTiter-GloTM Luminescent Cell Viability Assay reagent (Promega Japan, Tokyo, Japan) was added to evaluate intracellular ATP concentration according to the instruction.

Antimalarial activity of isolated compounds was evaluated according to the previous report [46]. The SYBR green assay is widely used for the antimalarial assay. This dye binds to parasite DNA and emits fluorescence reflecting the number of parasites in the red blood cells. In brief, 100 µL of P. falciparum parasite culture [47] was plated in a 96-well plate with various concentrations of the compounds. After incubation for 72 h at 37 °C in a humidified chamber under a gas mixture of 90%N_2_, 5%O_2_ and 5%CO_2_, the parasitemia was determined by SYBR Green I assay (Lonza Ltd., Basel, Switzerland) with a microplate reader at 485 and 530 nm. Dihydroartmesinin was used as the positive control and DMSO as a negative control. Human erythrocytes and plasma were obtained from the Nagasaki Red Cross Blood Center, and their usage was approved by the ethical committee of the Institute of Tropical Medicine, Nagasaki University.

The biological assays were performed three times for each assay and the IC_50_ values were calculated by linear regression using Microsoft Excel software.

## 4. Conclusions

The ethyl acetate extract of *A. galanga* was originally inactive against *L. major*. However, the chemical conversion by epoxidation generated significant activity to this parasite. The HPLC profile clearly indicated the constituent difference between the original and chemically converted extract (CCE) of *A. galanga*. Through the intensive fractionation and purification of CCE, six new phenylpropanoids (**1**–**6**) and four known compounds were obtained as active principles. The most active compound was hydroquinone (**7**), which further supports the importance of this chemical motif as shown in our previous report on the discovery of new anti-Leishmanial benzoquinones, named ardisiaquinones A–H [31]. In addition, the new phenylpropanoid **1** was revealed to inhibit two African *Trypanosoma* parasites effectively with high selective manner compared to *L. major*. Phenylpropanoids are an important constituent of plant essential oils. Several plant oils have been reported to show antiparasitic activity by inhibiting proteinase, dehydrogenase, and lipid trafficking [48]. The phenylpropanoids **1**–**6** isolated in this study may have a similar mode of action. The chemical synthesis of analogous compounds of **1** and the mechanism analysis are under investigation.

## Data Availability

Data is contained within the article or Appendix A.

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
