# Peer review of "Six New Phenylpropanoid Derivatives from Chemically Converted Extract of Alpinia galanga (L.) and Their Antiparasitic Activities"

_molecules, 2021, doi:10.3390/molecules26061756_

Round 1

Reviewer 1 Report

This paper describes the isolation of some remarkably simple compounds with anti parasitic activity against Leishmania and Trypanosoma. The scientific approach used is not new but the discovery of some new compounds and the demonstration of biological activity merit publication. Although the authors have provided extensive numerical characterisation data, there is no visible evidence of the purity of the samples evaluated. In my opinion analytical HPLC and NMR data must be provided in chart form in order to establish the purity of the compounds. This information could happily be presented as supplementary information but it is important for the validity of the work. I understand from the text that further work is in progress to define the mechanism of action of the new compounds, especially compound 1, and this is reasonable. But the paper lacks any comment on the biological activity of 4-hydroxyphenylpropanols and related compounds. From a quick look at the literature there is not very much in closely related compounds but some further evaluation of the results in the context of similar compounds would certainly improve the paper.  A few lines of text and a few references would suffice. On balance, therefore, I recommend publication subject to minor revision.

Reviewer 2 Report

The authors studied the effects of derivative compounds from Alpinia galanga and its anti-parasitic activities.  Although the idea is interesting, and the manuscript is easy to read, most part of the manuscript is related to the elucidation of the compounds. Regarding the anti-parasitic effect, the manuscript need more information, both for the methodology (i.e. how was checke the counts of parasites or the cultures) used to check this effect, and also for the discusion there are no discussion comparing the efect with that of other essential oils or other epoxides, there are some  studies regarding the effect of essential oils epoxides against protozoa. However the authors does not cite enough references in subheader 2.2., they only indicate that differences between trypanosoma and Plasmodium is due to belong to diferent phylum as well as the importance of hydroquinone reported previously in the scientific literature. The authors must improve this section. 

The authors also indicate several values with incertidumbre indication, but no statistical methods are indicated. Likewise, no reference to the number of assays for anti parasitic studies are specified.

Reviewer 3 Report

The authors report an extraction of a root to obtain biology active compounds.  The approach involves multiple extraction solvents followed by derivitization by epoxidation.  Hydrolysis leads to products.

I am not an expert in natural product extraction, so my questions are mostly about the whole extraction process.  In this work, they extracted the root with methanol, followed by EtOAc partitioning and epoxidation.  OK, I can accept their steps as written, but I am working on understanding things better.

1) the methanol extraction is a room temperature extraction.  What would happen at high temperatures?  It seems that this would lead to more extraction.  Then again, maybe that's not desired.  It's just not clear to me

2) I'm trying to understand physically how the rhizome was extracted.  Is it just that it is dropped in 10L of solvent?  How long does it stay there?  Is it shaken or stirred or agitated in any way?  And how is the rhizome removed from the solvent before it is evaporated to get the residue?

3) How is the aqueous suspension of the residue created?  Again, is there any agitation involved?

4) I am not familiar the authors' use of "partitioning" with ethyl acetate (I asked colleagues and they also not familiar with it either). Is that an extraction of the aqueous layer?

The original extract is not active in the bio-assay.  However, in the next step, they use dioxirane, an epoxidation reagent.  This makes the mixture biologically active.  They isolated 6 new compounds and 4 known compounds.  The most biologically active product is hydroquinone, but their compound 1 has some biological activity, so they conclude this is motiff to pursue.

Now, in light of the introduction, there is a major problem with this study.   Starting in line 82, the authors talk about the rhizome as a treatment for various ailments, and refer to the pharmocological activities, culminating in line 86, which says "there is no report about its activity as an anti-parasitic agent." (They immediately cite a report about it being ineffective against lmm - is lmm not a parasite?)

In this respect, the conclusion of this work is that there is no activity of the a. galanga rhizome extract.  And, to be fair, that is the first sentence of the conclusion section.  And so all of the positive activities reported in the paper are irrespective of that actual natural product.

The introduction gives the impression that this is about the extraction of active compounds from natural sources.  It's not.  As the authors indicate, the structures they isolate are not present in the original extraction.  Therefore, while the A. galanga extract provided a source of starting scaffolds for the products obtained in this work, it's not really the premise of the study, which is more that you need chemical modification step to make this stuff worth anything.

I realize that the goal of the introduction is to provide some perspective for the work, but it's deceptive and it concerns me.  This work is not about the pharmocological properties of A. galanga. The paragraph starting at line 65 is fair.  The next paragraph is not as much.

Reviewer 4 Report

In this manuscript the authors have reported 6 new Phenylpropanoid Derivatives from Chemically Converted Extract of Alpinia galanga and elucidated their structures using classic spectroscopic studies, and Investigated these compounds for their antiprotozoal activity.

Here are my comments:

1) Please correct the manuscript title to Six New Phenylpropanoid Derivatives from Chemically Converted Extract of Alpinia galanga (L.) and their anti-parasitic activities.

2) Please assign the numbering for at least for one structure, so that it will be easy for the readers to understand which Carbon or Proton you are referring to.

3) If possible, include the structures of compounds 7, 8, 9, 10 and also the positive controls used in the study at respective places.

4) Label H-H COSY and HMBC correlations of compounds with different colors. (it is really confusing and hard to see as it is presented). Also, please include a Table for the COSY, and HMBC correlations.

5) On Page 6, Line 195: Please assign H-7 and H-8 you are referring to in compound 6. I do not see H-7 and H-8 in any of these structures. (Am I missing something here).

6) Page 7, Lines 208 to 210: The sentence is confusing to the readers. Please rephrase the sentence.

7) Page 8, Lines 241 - 247: Experimental procedure for CCE from E is confusing. Please rephrase the sentences.

8) It will be a great value to this manuscript, if authors can add the sprectral data as supporting information attachment.

9) English in this manuscript is fine except in few paragraphs, where authors used lengthy sentences and no punctuations. For example, Page 2, Lines 68 - 62.

To conclude, given the originality, and broader interest in the scientific community I recommend the editors to accept and publish this manuscript with moderate changes as mentioned in comments 1 to 9.
